# Evaluating the Cysteine-Rich and Catalytic Subdomains of Human Tyrosinase and OCA1-Related Mutants Using 1 μs Molecular Dynamics Simulation

**DOI:** 10.3390/ijms241713032

**Published:** 2023-08-22

**Authors:** Taariq Woods, Yuri V. Sergeev

**Affiliations:** National Eye Institute, National Institutes of Health, Bethesda, MD 20892, USA; woods.taariq@gmail.com

**Keywords:** human tyrosinase, homology model, molecular dynamics simulation of protein urea unfolding, molten globule, cysteine-rich and tyrosinase subdomains, OCA1

## Abstract

The inherited disorder oculocutaneous albinism type 1 (OCA1) is caused by mutations in the *TYR* gene encoding tyrosinase (Tyr), an enzyme essential to producing pigments throughout the human body. The intramelanosomal domain of Tyr consists of the cysteine-rich and tyrosinase catalytic subdomains, which are essential for enzymatic activity. In protein unfolding, the roles of these subdomains are not well established. Here, we performed six molecular dynamics simulations at room temperature for Tyr and OCA1-related mutant variants P406L and R402Q intramelanosomal domains. The proteins were simulated for 1 μs in water and urea to induce unfolding. In urea, we observed increases in surface area, decreases in intramolecular hydrogen bonding, and decreases in hydrophobic interactions, suggesting a ‘molten globule’ state for each protein. Between all conditions, the cysteine-rich subdomain remains stable, whereas the catalytic subdomain shows increased flexibility. This flexibility is intensified by the P406L mutation, while R402Q increases the catalytic domain’s rigidity. The cysteine-rich subdomain is rigid, preventing the protein from unfolding, whereas the flexibility of the catalytic subdomain accommodates mutational changes that could inhibit activity. These findings match the conclusions from our experimental work suggesting the function alteration by the P406L mutation, and the potential role of R402Q as a polymorphism.

## 1. Introduction

Oculocutaneous albinism (OCA) is an autosomal recessive disease caused by perturbations in the melanogenesis pathway that causes a broad range of pathologies including but not limited to nystagmus, hypopigmentation, and foveal hypoplasia [1]. This disease is separated into eight subtypes, with each subtype being classified by mutations in a different gene. These subtypes are named OCA type 1–8 and are associated with the following genes in respective order: *TYR*, *OCA2* (P gene), *TYRP1*, *SLC45A2*, *OCA5*, *SLC24A5*, *C10ORF11*, and *DCT* (*TYRP2*) (OMIM^®^, An Online Catalog of Human Genes and Genetic Disorders; http://www.omim.org; accessed on 3 May 2023). Among these subtypes, the most severe is oculocutaneous albinism type 1 (OCA1), caused by perturbations in the tyrosinase (Tyr) enzyme encoded by the *TYR* gene. Patients with this form of albinism can be further described as having OCA1A or OCA1B [2]. These forms are defined by differences in their severity, as OCA1A causes complete inhibition of melanin production while OCA1B causes a reduction in pigmentation of the hair, skin, and eyes. The phenotype of OCA1A is significant as it is the only OCA subtype to inhibit all levels of pigmentation, highlighting the essential role of Tyr. Over 300 mutations in this enzyme have been identified in OCA1 patients [3].

Tyr is a transmembrane metalloenzyme that is essential to produce melanin. The domain architecture of this enzyme consists of a signal peptide (1–23), cysteine-rich subdomain (24–118), catalytic tyrosinase domain (179–409), transmembrane helix (471–501), and cytoplasmic domain (502–519) (Appendix A). The intramelanosomal body of the enzyme encapsulates the cysteine-rich subdomain and tyrosinase domain, which is essential for enzymatic function. The maturation of this protein is dependent upon the glycosylation of seven asparagine residues, as the enzyme is folded in the endoplasmic reticulum and transported to the melanosome [4]. Once moved to the melanosome, Tyr remains attached to the organelle membrane where it catalyzes the oxidation of L-tyrosine and L-3,4-dihydroxyphenylalanine (L-DOPA) to dopaquinone [5]. This reaction is both the first and rate-limiting step in melanin synthesis, making it essential for producing all melanin pigments.

Our group has previously demonstrated that the recombinant intramelanosomal domain of human tyrosinase (Tyr_tr_) is required to retain enzymatic activity [6]. By removing the transmembrane and cytoplasmic regions of Tyr, Tyr_tr_ is also soluble in an aqueous solution, making it desirable for protein purification. Using this construct, we can move from genetic associations of disease-associated mutation to identifying changes in protein stability and activity that confer these phenotypic changes that occur in OCA1 patients. Two OCA1B-associated mutations (P406L, R402Q) have been shown to cause instability and decreased activity of Tyr_tr_ relative to the severity of pigmentation defects in vivo [7]. In vitro urea unfolding experiments have also offered insights into the presence of folded and unfolded states of these variant enzymes but have not brought clarity to the structural determinants of these differences in activity. Through computation, we hoped to investigate the underlying mechanisms behind the heterogenic activity of these hypomorphic mutations.

Homology modeling is a method that takes the amino acid sequence of a protein with an unknown structure and predicts this new structure based on similar solved structures [8]. This method was previously used to predict the structure of Tyr_tr_ [9]. Our homology model of Tyr_tr_ is composed of two subdomains. The first is an extended EGF-like domain containing a five-membered disulfide bridge network, defining the cysteine-rich subdomain. This network provides structural constraints to the globular domain, increasing its rigidity and stability. The Tyr catalytic subdomain houses a bundle of four embedded helices that make up the Tyr active site. This active site utilizes six histidine residues (H180, H202, H211, H363, H367, H390), the first three coordinating one copper ion (CuA), and the last three coordinating a second (CuB). These ions bind an O_2_ molecule to form the complete active site [10]. To gain information about structural differences in these domains between the wild type and mutants, we also explore the utility of long-duration molecular dynamics (MD) simulations in urea to assist protein destabilization [11]. At the molecular scale, protein unfolding is a large process that can occur on a timescale of seconds or longer [12]. Considering the current limitations in computation, and the expensive process of MD, which requires weeks to simulate moderately sized proteins, we aimed to simulate the destabilization of Tyr_tr_ in urea at the microsecond timescale.

## 2. Results

### 2.1. Tyrosinase Intramelanosomal Domain and Mutant Variants

Three main structures are essential for retaining the globular structure of Tyr_tr_. These structures include the disulfide bridge network located within the cysteine-rich subdomain, the amino acids making up the 4-alpha-helical bundle that forms the hydrophobic core of the catalytic subdomain, and the six histidine residues that hold the active site copper cluster. Interestingly, the P406L and R402Q mutations are present on the periphery of the Tyr catalytic domain (Figure 1). Not only do these mutants supply us with clinically relevant variants to help understand the structural determinants of Tyr stability, but they also provide examples of mutations in ‘structurally insignificant’ regions of the protein that indeed have broader effects on the enzyme. Using our unfolding mutation screen (UMS), we were able to predict the destabilizing effects of these mutations that align with their clinical significance (Table 1). At the very least, changes in a single amino acid lying at the periphery of a globular domain may introduce local changes in structural rigidity that cause global changes in stability and enzymatic activity. For the Tyr intramelanosomal domain, the N-terminus is embedded at the interface between the cysteine-rich and catalytic subdomains. 

### 2.2. 1 μs Molecular Dynamics

Also within this area is a loop that binds to the periphery of the globule to retain its stability. After 1 μs, we observed destabilization of this loop and a beta-sheet proximal to the active site copper clusters in both P406L and R402Q. Stabilizing contacts remain in both regions for the WT (Figure 2). Following the relative predicted severity of each mutation, these regions exhibit destabilization of their secondary structure and move further from the embedded N-terminus and active site, respectively. Active site copper clusters remained relatively stable between all conditions and mutations. Also, observe that a C-terminal portion of the intramelanosomal domain rests at the interface of its subdomains (Figure 3). We sought to understand the changes that occur in these destabilizing conditions for both the subdomains and non-subdomain regions of Tyr.

Although 1 μs of MD is simply too short of a timespan to observe a complete picture of urea-induced protein unfolding, we do glimpse the transition of the fully folded domain to a state different from the native state. This transition is most clearly observed when comparing the behaviors of these structures in water and 8M urea.

### 2.3. Molten Globule-like State

Structural differences in native and 1 μs transitioned states were determined using the average solvent-accessible surface area (SASA) and solvent-accessible volume of the protein at each nanosecond of the simulations. We observed that all three Tyr intramelanosomal domains experience secondary structural expansion throughout the MD in the 8M urea environments (Figure 4). Across all structures, the water and urea conditions form two distinct trajectories in which Tyr continually expands in urea while becoming more compact (WT, R402Q) or remaining relatively stable (P406L) in water. Interestingly, R402Q retains the lowest surface area in 8M urea compared to WT and P406L. Also, the WT and R402Q have seemingly identical trajectories in water, whereas the WT and P406L have overlapping trajectories in 8M urea. Moreover, we observed clear deviations in the severity of destabilization between the WT and each mutant variant, indicating that the structure achieves the transitioned state at 1 μs, which could be characterized as a ‘molten globule’-like state.

Beyond our whole-protein SASA and volume analyses, we calculated cavity volumes to identify specific structural components that account for the global expansions we observed in 8M urea. Interestingly, average cavity volumes were highest for the WT structure and lowest for P406L (Table 2). This result is contradictory to our initial expectations that structural instability would be positively associated with cavity volume. However, one possible explanation for this decrease is described in our discussion of these findings (internal expansion/‘rounding’ of globular structure). It may also be important to note that P406L has the most variable cavity volumes in water. The variability we observe supports our prediction that the substitution of proline decreases overall structural stability. Additionally, R402Q had the least variable cavity volume in water, further suggesting that this mutation causes an increase in the structural rigidity of the globule.

### 2.4. Interatomic Hydrophobic Contacts

We hypothesized that proline 406 plays an important role in ensuring structural rigidity of the Tyr catalytic subdomain. To test this role, we calculated the number of inter-atomic hydrophobic contacts P406 forms between neighboring residues (Figure 5, Panels A–E). WT in water forms 12 contacts between 5 residues. P406L in water forms 16 contacts between 3 residues (Panel B). R402Q in water forms 19 contacts between 7 residues (Panel C). This structure also experiences lengthening of the core alpha-helix that houses residue 402. WT in 8M urea forms eight contacts between three residues, with one contacting residue (Y467) present at the flexible C-terminus (Panel D). P406L in 8M urea forms five contacts between three residues (Panel E). R402Q in 8M urea forms 14 contacts between 5 residues (Panel F). In summary, at 1 μs, P406 is embedded within the catalytic domain of the WT and forms 12 contacts between 5 residues. Three of these residues are members of the core helix that also houses R402. In 8M urea, we observed a decrease in the number of interatomic contacts and contacting residues. These trends are observed between all structures; however, the P406L substitution causes a local decrease in structural rigidity as the contacts L406 makes are present on the solvent-exposed side of the protein. Contrarily, the R402Q mutation results in a strengthened hydrophobic contact network around P406 via the lengthening of the adjacent core helix. Overall, we observed that the P406L and R402Q mutations have opposite influences on the local hydrophobic contact network, causing a respective increase and decrease of flexibility in the periphery of the Tyr catalytic subdomain.

### 2.5. Hydrogen Bonds

In addition to hydrophobic contacts, intramolecular hydrogen bonds play a pivotal role in stabilizing the secondary structure of proteins. By calculating the total number of hydrogen bonds at each nanosecond, we gained a view of the changes in secondary structure stability in each variant throughout the MD simulations (Figure 6, Panels A–F).

From this analysis, we see that the simulation of Tyr_tr_ in water naturally stabilizes the protein structure by increasing the total number of intramolecular hydrogen bonds. The formation of these bonds is stifled only for the P406L variant, demonstrating its destabilizing nature. Contrarily, the 8M urea condition causes the breakage of hydrogen bonds in all variants. The most drastic decrease in these interactions occurs for the P406L mutation, with R402Q experiencing resistance to destabilization relative to the WT.

When summarizing our hydrogen bonding data by fitting it to a linear curve, we observe clearer influences of each mutation on the overall stability of the Tyr intramelanosomal domain (Table 3). For the WT, the number of hydrogen bonds is expected to increase by 13 throughout 1 μs in water, whereas the structure experiences a decrease of 9 bonds in 8M urea. These changes are expected to be quite similar between the WT and R402Q variants, with an improvement in stability by one more hydrogen bond formed in water and three hydrogen bonds conserved in 8M urea for R402Q. P406L benefits the least from simulation in water with a predicted increase of eight bonds and breaks nine more bonds than the WT in urea. These data provide us a clear view of the global effect of each mutation on the Tyr_tr_ structure, as the P406L mutation inhibits the formation of stabilizing intramolecular hydrogen bonds, whereas the R402Q mutation prevents the breakage of these bonds.

### 2.6. Electrostatics

Finally, we evaluated the strength of ionic interactions within each structure to gauge the effect of urea on electrostatic interactions. For this analysis, we focused on the subdomain interface, as this area of the globule forms a cleft that is most easily accessible by solvent compared to the more tightly packed subdomains themselves. We observed that urea can occupy the space between the embedded C-terminal region of the intramelanosomal domain and the cysteine-rich subdomain of Tyr (Figure 7). The space between the cysteine-rich and catalytic subdomains forms a cleft that is highly solvent-accessible. Urea can fill the gap between the interface residues and the disulfide bridge network. With the proposed ability of urea to inhibit interactions between charged residues, the solvent within this cleft may contribute to the molten globule by destabilizing electrostatic interactions within this region [13]. Given this fact, we focused on this fold to evaluate the impact of urea on these stabilizing forces. We found that both these structures experience transient ionic interactions in this fold that are perturbed in 8M urea (Figure 8). The WT shows somewhat stable ionic interactions in water throughout 1 μs, which are weakened between 200 and 400 ns, then increase in strength beyond 500 ns of simulation in urea. Ionic interactions for R402Q are inhibited to a greater extent compared to the WT in urea, although the total energy of these interactions is decreased for R402Q in water as well. Interestingly, P406L shows similar ionic interaction profiles in water and urea, potentially due to the increase in internal volume of the globule relative to the WT and R402Q. Each variant is shown to have a distinct effect on stabilizing electrostatic interactions within the subdomain interface, with R402Q inhibiting the formation of these bridges in all conditions while P406L prevents the breakage of these interactions.

### 2.7. RMSD and RMSF Plots

Calculations for RMSDs and Cα-RMSFs were performed using YASARA, as described in the Methods section. RMSDs were generated by superposing each molecular dynamics snapshot to their Cα atoms to the original structure, calculated, and plotted in Figure 9 (Panel A). These superposed structures were saved as PDB files for the subsequent RMSF analysis and plotted in Figure 9 (Panel B). Interestingly, the RMSD values showed the highest deviations in P406L in water and the wild type in 8M urea. Similar to our SASA calculations, the R402Q structure retains its structure better than all other proteins. During the MD runs, the most significant fluctuations were observed for the loop formed by residues 150–170 and for the C-terminus located above the residue 450. Contrary to the RMSD values, there is much overlap between RMSF peaks across all structures with only the wild type in water displaying reduced fluctuations. These reduced fluctuations are observed at the peaks between residues 50–75, 150–170, and to a lesser extent 300–315 and 400–420.

## 3. Discussion

Tyr is responsible for catalyzing the rate-limiting step in the production of melanin pigments. Genetic perturbations in this enzyme can cause partial or complete inhibition of melanogenesis by destabilizing the native folds of this domain. Clinically, these perturbations manifest as the autosomal recessive disease known as OCA1. This disease has two subtypes, defined as OCA1A and OCA1B. Here, we investigated the mechanisms by which the OCA1B mutations P406L and R402Q destabilize Tyr_tr_ and the specific effects they have on each subdomain of this enzyme. To accomplish this, we built models of each variant from our previously published homology model and simulated these structures for 1 μs of MD in water and 8M urea. Using these simulations in urea, we accelerated the timescale of protein unfolding enough to obtain a glimpse into the molten globule formation of each enzyme variant. With this feat achieved, we were able to observe the unique structural differences between Tyr_tr_ and its mutants to mechanistically describe the destabilizing effects of these amino acid substitutions at the atomic scale. This information is crucial for the downstream development of targeted therapies, which may recover function in these hypomorphic Tyr mutations.

Given the essential role of Tyr in the melanogenesis pathway, it is not surprising that there are over 300 known mutations in the *TYR* gene that are associated with OCA1. Studying deleterious mutations, like that of OCA1A-associated mutations, may be more intuitive, as many of these gene products do not reach full maturity and are degraded by the human proteasome, or simply cannot produce a natively folded protein [7]. In contrast, hypomorphic mutations such as P406L and R402Q allow us to study the roles each of the native amino acids play in the overall stability and activity of these enzymes. With P406 and R402 being located at the periphery of the catalytic domain of Tyr_tr_, structural observations alone cannot offer us insights into the roles of these residues. MD simulations function to fill this gap by providing us information on how Tyr_tr_ behaves differently with each substitution. In addition, simulations in 8M urea allow us to compare and accelerate the timescale at which larger structural changes would occur beyond what is computationally possible to simulate under these conditions.

From this long-duration MD simulation, we can capture events along the unfolding pathway in fine detail. There are four distinct stages in the protein unfolding pathway: the natively folded protein, molten globule, premolten globule, and unfolded protein [14]. These stages are conserved for both folding and unfolding processes. This framework defines the unique structural identities of each stage in protein folding, providing us a good general sense of the types of protein movements that are significant in the folding process. However, it neglects to pinpoint how the protein transitions from one stage to another. The identification of these transition states is a problem that MD simulation excels at helping us solve [15]. For Tyr_tr_, we can observe a broad range of structural changes that occur before and during equilibration, compaction, and expansion of the globule in different conditions. These subtle structure shifts are difficult, if not impossible to observe in a three-dimensional protein structure in many cases. Studying these changes allows us to target specific motifs within proteins that contribute to instability in natural and mutated states. These motifs may act as targets for future studies aiming to restore function to OCA1-related enzyme variants.

All-atom MD at the μs timescale offers a nuanced view of protein motions that can help us understand structural determinants of protein stability and activity; however, this method is not without its challenges. With our MD configuration, the three-dimensional motion of each atom is calculated based on intrinsic energy characteristics and covalent and noncovalent interactions it formed with other atoms at every femtosecond of simulated time [16]. Naturally, this method requires massive computational expenditure, which increases with the size of the simulated system. Increasing the timestep, which is the interval at which motions are calculated, reduces the time to solution at the cost of decreasing accuracy [17]. However, increasing the speed of a simulation without augmenting experimental parameters is possible. Beyond utilizing high-performance compute nodes with powerful graphical processing units (GPUs), it is possible to parallelize a simulation between multiple compute nodes using a message-passing interface [18,19]. Unfortunately, our simulation package does not support internode communication for this level of optimization. For these reasons, we were limited to the μs timescale for our simulations (Appendix A).

Despite these computational restraints, we were able to simulate the molten globule transition of Tyr_tr_ and OCA1-related variants using MD in 8M urea. Molten globule formation is defined by the expansion of the protein and the unpacking of mobile sidechains without destabilization of the native secondary structure [14]. In all 8M urea conditions, we observed increases in SASA and solvent-accessible volumes (Appendix A). In general, predictions for the changes in P406L were intuitive. The severity of expansion was highest over time for P406L, caused by a decrease in hydrophobic interactions local to the catalytic domain, and a global decrease in hydrogen bonds within the globule. Interestingly, P406L experienced no significant decrease in ionic interactions in the subdomain interface and had the lowest total cavity volumes among all proteins. These results may concern each other, as P406L is shown to have the least stable structure even in water. It is important to note that this structure experiences the largest standard deviations in cavity volume, potentially relating to increased flexibility in sidechains and loops throughout the protein. As P406L expands, the protein’s internal volume may increase as the volume of native cavities on the globule periphery decrease. In essence, the globular structure grows like that of a balloon before more solvent is allowed to enter the protein, triggering the transition from the molten globule to the premolten globule [14]. The cysteine-rich subdomain does not appear to undergo many, if any, changes for this structure under any condition, whereas the catalytic domain experiences a global increase in structural flexibility reminiscent of the molten globule. Contrary to the P406L mutation, R402Q has rather intricate effects on the entire intramelanosomal domain. Relative to the WT, R402Q is less vulnerable to urea-induced expansion, although it still occurs at a slower rate. We see that R402 is located at the peripheral end of one of the α-helical bundles of the Tyr catalytic domain. When this relatively large amino acid is substituted for glutamine, this appears to increase the flexibility of the local secondary structure enough to extend the helix by one amino acid (H404). This causes local compaction of the helix and adjacent sidechain, increasing the hydrophobic interactions in this area. However, this increased rigidity in the catalytic domain seems to have negative effects on the subdomain interface, as ionic interactions are decreased both in water and urea. It is known that globular proteins often adopt a rigid active site confirmation, which is accompanied by decreases in rigidity in the globule periphery. With additional rigidity in the Tyr_tr_ periphery from this mutation, it may decrease rigidity in other regions, such as the subdomain interface and cysteine-rich domain, causing a decrease in ionic interactions. Likewise, decreases in rigidity in the catalytic subdomain periphery may increase the rigidity in the subdomain interface, resulting in ionic interactions being sustained in 8M urea for the P406L variant. Indeed, the interplay between local and global changes in protein rigidity for Tyr_tr_ should be further studied.

Overall, we were able to simulate just a portion of the protein unfolding curve for Tyr_tr_, but these data provide us valuable insight into the dynamic transition states in the unfolding pathway. It is known that protein folding/unfolding can occur on the timescale of microseconds to seconds. However, the folding timescale varies based on the size and complexity of the molecule [12]. Relative to the sigmoidal protein unfolding curve, we expect that our simulation has only reached the very beginning of the upward slope toward full unfolding [7]. We also predicted that the molten globule forms specifically within the Tyr catalytic subdomain and subdomain interface of this enzyme. In the future, we hope to validate not only the existence of this molten globule in the catalytic subdomain but also the real timescale for these unfolding transitions as well. The use of circular dichroism (CD) and solution nuclear magnetic resonance (NMR) spectroscopy to identify the molten globular states of proteins has been well studied [20]. This task is partly complicated by the nature of Tyr_tr_, as it is not only held together via noncovalent interactions but also with seven total disulfide bonds and a copper–oxygen cluster within the active site. However, these methods may allow us to identify the stages of unfolding for Tyr_tr_, complementing the information we gather about the unfolding transition states from our MD studies.

## 4. Materials and Methods

### 4.1. Molecular Modeling—WT and Mutants

The Tyrtr homology model was obtained using the structure of homologous protein Tyrp1 (PDB: 5M8L) and accessible from NEI Commons Ocular Proteome Database (https://neicommons.nei.nih.gov/#/proteomeData, accessed on 1 May 2022). We removed the signal peptide region in UCSF Chimera and extended the C-terminal truncation by 20 residues with the Terminal Extension function within YASARA’s homology modeling package (YASARA Biosciences GmbH, Vienna, Austria, EU). The tyrosinase domain was glycosylated in positions as described previously [9].

The bonds coordinating copper atoms in the active site were replaced with pseudo bonds maintaining their specified bond lengths and angles throughout the simulations. Charges for the Cu ions were automatically assigned by the YASARA AutoSMILES program. Active site copper cluster geometries were optimized by placing both copper ions 3 Å away from bonded histidine residues, then using semiempirical quantum mechanics via MOPAC 7 [21]. The implicit solvent was added for the optimization using the COSMO model [22]. The resulting model was validated using SAVES (https://www.doe-mbi.ucla.edu/services/; accessed on 3 May 2023). Two OCA1-related mutants (P406L and R402Q) were built from our validated Tyr model using the Edit > Swap > Residue function in YASARA.

### 4.2. Computer Simulations

For each protein structure, we created a simulation cubic cell of 84 Å × 84 Å × 84 Å extending 10 Å beyond all protein atoms and filled the cell with pure water or aqueous urea using the TIP3P water model. The urea solvent model was built from the 3D conformer SDF file of a single urea molecule from PubChem (https://pubchem.ncbi.nlm.nih.gov/compound/Urea, accessed on 2 June 2022). The 3D conformer was energy minimized and saved as a PDB file in Chimera then loaded into YASARA. The urea model was parameterized using the AutoSMILES program, creating the final model used for all solvent environments (Appendix A). The energy-minimized urea molecule was placed into the empty simulation cell and used to fill the cell. Aqueous urea solvents were generated as described in the documentation of YASARA’s FillCellObj function, in which the concentrations of each solvent component are determined separately using a weighted density value as shown below in Equations (1)–(4).
(1)$$\frac{m_{urea}}{V_{Total}}=c_{urea}\times M_{urea}$$
(2)$$V_{urea}=\frac{m_{urea}}{{\rho {}^{\circ}}_{urea}}$$
(3)$$\rho_{urea}=\frac{V_{urea}}{V_{Total}}\times {\rho {}^{\circ}}_{urea}$$
(4)$$\rho_{water}=(1-\frac{V_{urea}}{V_{Total}})\times {\rho {}^{\circ}}_{water}$$

Here, *m_urea_* and *V_urea_* are the mass and volume of urea within the simulation cell. *ρ_urea_* and *ρ_water_* are the standard density of urea and water (1.32 g/cm^3^, 1 g/cm^3^). *ρ_urea_* and *ρ_water_* are the densities of urea and water within the simulation cell at the target molarity. *V_total_* is the total volume within the simulation cell. *M_urea and_ c_urea_* are the molar mass and molarity of urea in the simulation.

After the solvents were created and energy minimized, the Tyr structure was placed in the center of the simulation cell along with a mass fraction of 0.9% NaCl to neutralize the protein. Solvent environments were energy minimized before each protein structure was added, and the completed simulation system was energy-minimized again before beginning each MD experiment. All solvent molecules within 2 Å of the protein were removed to avoid bumps. Protein equilibration was evaluated using the changes in the accessible surface of each globule. This was shown by sharp changes in surface area values followed by a flattening curve showing that the proteins had adjusted to the simulation environments. This flattening curve was observed in all structures within the first 100 ns of the simulations. The cell neutralization and pKa prediction script within YASARA was used to predict pKa values as a function of electrostatic potentials calculated using Ewald summation, hydrogen bonds, and accessible surface areas of titratable groups.

The resulting systems were energy-minimized before running 1 ms of MD using a modified version of the ‘md_run.mcr’ macro in YASARA. This macro ran MD with the AMBER14 forcefield, with intermolecular forces calculated every 2 fs and intramolecular forces calculated every 1 fs [23]. Since the default pressure control algorithm is optimized for pure water solvents, and we are approximating the aqueous urea solvent densities, we utilized the built-in Manometer1D pressure control system for all experiments within our modified molecular dynamics script in YASARA. Simulation trajectories were saved every 0.1 ns for subsequent analysis. Simulations for all proteins were run in triplicate using a different random seed for each condition. All simulations were performed on the NIH Biowulf high-performing computation cluster.

### 4.3. Protein Structure Visualization and Comparisons

All structure alignments and domain visualizations were created in UCSF Chimera. The structures of Tyr_tr_, P406L, and R406Q were aligned after 1 μs of MD using the MatchMaker tool. The WT was used as the reference and the best-aligning pairs of chains between the reference and match structures were identified using the Needleman–Wunsch algorithm and the blocks substitution matrix (BLOSUM-62). The gap extension penalty was set to 1 and a secondary structure score of 30% was included. Secondary structure assignments were computed for each model, and long atom pairs were matched using iterative pruning until no pair exceeded 2.0 Å. Tyr_tr_ subdomains and the subdomain interface region were identified and compared using the Sequence tool. Embedded solvent visualizations were created in YASARA. This was achieved by loading the MD simulation file of the WT in 8M urea at 1 μs into YASARA, turning off the simulation, and deleting all solvent molecules further than 21 Å from the protein center of mass. To visualize solvent embedded in the subdomain interface, all solvent molecules further than 5 Å from the residues 410–450 (interface region) were removed and the resulting simulation was saved as a PDB file for visualization in UCSF Chimera.

### 4.4. Unfolding Mutation Screen (UMS)

To further investigate how mutations at these positions of Tyr_tr_ may lead to destabilization, we utilized our global computational mutagenesis pipeline. This pipeline is a collection of programs that allow the user to model protein structures and calculate the destabilizing effects of missense mutations. One of these programs, the unfolding mutation screen (UMS), calculates unfolding propensities from free energy change (ΔΔG) values for all possible missense variants of the structure. Unfolding propensities for each residue position are combined to generate a foldability score ranging from 0 to 19, which indicates the likelihood of a mutation at a certain position resulting in protein unfolding. These foldability scores are mapped to their corresponding residue positions on the structure, providing us with a three-dimensional view of which positions are critical for stability.

### 4.5. Solvent-Accessible Surface Areas (SASA) and Cavity Volumes

SASA values were calculated for Tyr_tr_ and mutant variants in water and 8M urea for every nanosecond of simulation time using the SurfObj function in YASARA. MOLE 2.5 was used to find the cavity and void volumes of each structure at 0M and 8M urea at 1 μs. This cavity analysis generates an interface that displays each cavity along with its associated volume. These volumes were summed, and the average total cavity volume and standard deviations were compared between structures.

### 4.6. Intramolecular Noncovalent Interactions

Hydrogen bonding interactions were calculated for each entire protein structure at every nanosecond of the simulations using the ListHboAtom function in YASARA and defining the atom range as the entire protein object. To calculate interatomic hydrophobic contacts with residue 406, all atoms of this residue were selected and designated using the Find Clashes/Contacts tool in UCSF Chimera. Once the atoms are designated and all other atoms are present in the structure, the contact parameters were defined to identify atom pairs with a van der Waals radius overlap less than 0 and greater than −0.4 Å. Intraresidue contacts and contact pairs of 4 or fewer bonds apart were ignored, and intramolecule contacts were included. Ionic interaction energies were derived from the ListInt function in YASARA for specified residue groups within Tyr_tr_ and variants. The distributions of ionic interaction energies were plotted using the seaborn package in Python.

### 4.7. Measures for MD Quality Assessment

The RMSD of an ensemble of structures from a reference structure was calculated using the RMSD command in YASARA and additional inhouse Python scripts. The ensemble of molecular dynamics snapshots of 1 ns interval was aligned with the reference structure, based on a set of selected Cα atoms, followed by calculations of RMSD values between corresponding atoms of the snapshots and reference structure.

The RMSF was generated using the superimposed molecular dynamics snapshots for every 25 ns interval. These superposed structures were saved as PDB files with their updated coordinates to remove all translational and rotational movements. The RMSF analysis was carried out by loading each superposed structure into YASARA, adding the Cα-atom positions to a calculation table using the AddPosAtom function to calculate the averages and standard deviations using the built-in RMSF command.

## 5. Conclusions

In this work, we simulated environments containing 8M urea to analyze the destabilization pathways of Tyr_tr_ and OCA1B-related mutant variants P406L and R402Q. In this condition, we observed structural changes indicative of a molten globular state, including increases in surface area, the breakage of intramolecular hydrogen bonds, and decreased hydrophobic interactions localized to the mutation sites. These changes are localized to the catalytic subdomain of Tyr_tr_, causing increases in structural flexibility proximal to the active site while keeping its embedded copper cluster structurally intact. Between all our simulations, the cysteine-rich subdomain experienced minimal fluctuation and remained rigid. The structural roles of these subdomains are essential, as the catalytic domain allows for modulation of the active site space without complete inhibition of activity, and the cysteine-rich subdomain prevents the protein from experiencing severe unfolding. Particularly, to our OCA1-related mutations, P406L causes further decreases in the rigidity of the catalytic subdomain, hastening the molten globule transition in urea. R402Q introduces local increases in rigidity that help the protein resist urea-induced destabilization at the potential cost of adherent active site modifications. These results are consistent with the conclusions from our experimental work on these mutants, which shows the more severe nature of the P406L mutation and the potential role of R402Q as a polymorphism. This evaluation of Tyr provides us structural information about OCA1-related gene products, helping us connect genotypic risk factors for disease with clinical phenotypes for patients. These studies may aid future developments for targeted therapies to restore function in Tyr and other important enzymes in the melanogenic pathway.

## Figures and Tables

**Figure 1 ijms-24-13032-f001:**
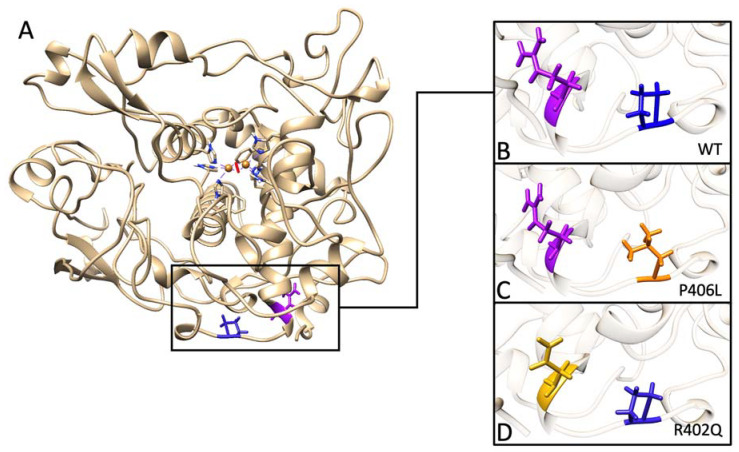
Homology model of Tyr intramelanosomal domain and the changes in the models of mutant variants. Panel (**A**): The homology model of the human Tyr intramelanosomal domain is shown by a beige ribbon structure. Within the active site, six histidine residues coordinate two copper atoms, which are shown in the center. The area of protein affected by the mutations is included in a black rectangular frame. The model was taken and optimized from our NEI Commons Ocular Proteome database (https://neicommons.nei.nih.gov/#/proteomeData/, accessed on 1 May 2022). Panel (**B**): Area of wild-type protein affected by mutations. The location of affected residues, arginine 402 (purple) and proline 406 (blue) in wild-type Tyr (WT). Panel (**C**): The P406L mutation is highlighted in orange. Panel (**D**): In the R402Q mutant variant, the change is shown in yellow.

**Figure 2 ijms-24-13032-f002:**
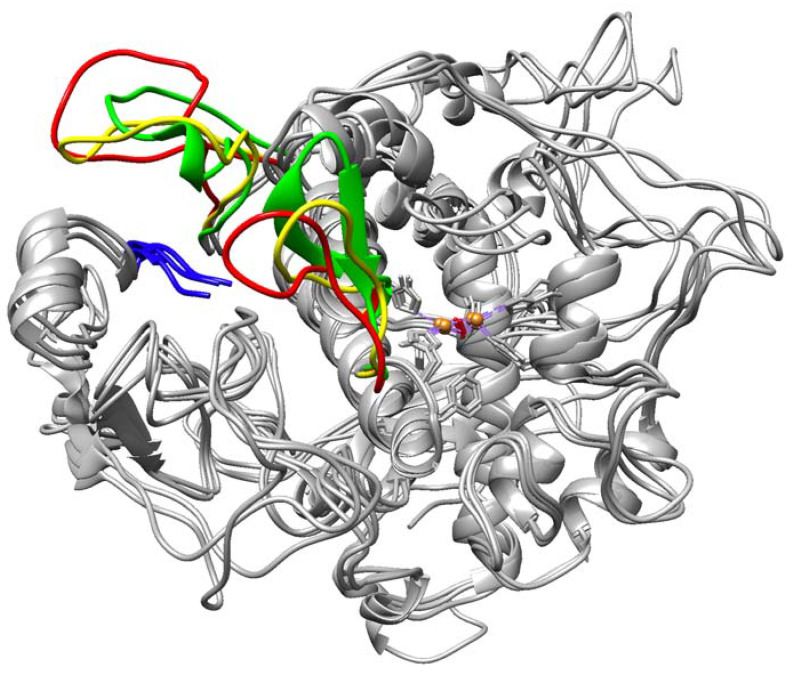
Destabilization of secondary structure proximal to N-termini and active site in Tyr mutant variants. Areas of focus are colored as follows: Wild-type of Tyr (green) and two mutant variants P406L (red) and R402Q (yellow) structures were visualized and aligned at 1 μs. N-termini are labeled blue and active site copper atoms are labeled brown. Bonds between copper atoms and coordinating histidine residues are present as dashed purple lines. The rest of the structures with fewer changes are colored grey.

**Figure 3 ijms-24-13032-f003:**
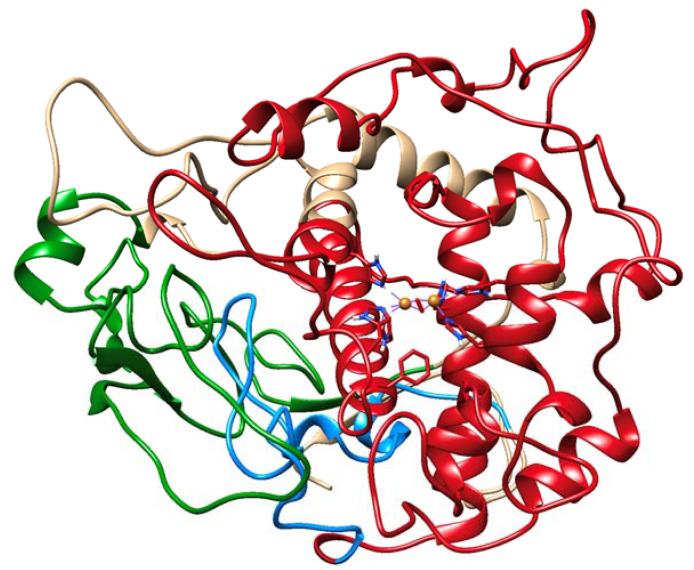
Intramelanosomal domain of Tyr: subdomains and non-subdomain regions. The active site of Tyr lies within the catalytic Tyr subdomain (red). The cysteine-rich subdomain rests on the periphery of the globule (green). The residues beyond the catalytic region make up the subdomain interface (blue).

**Figure 4 ijms-24-13032-f004:**
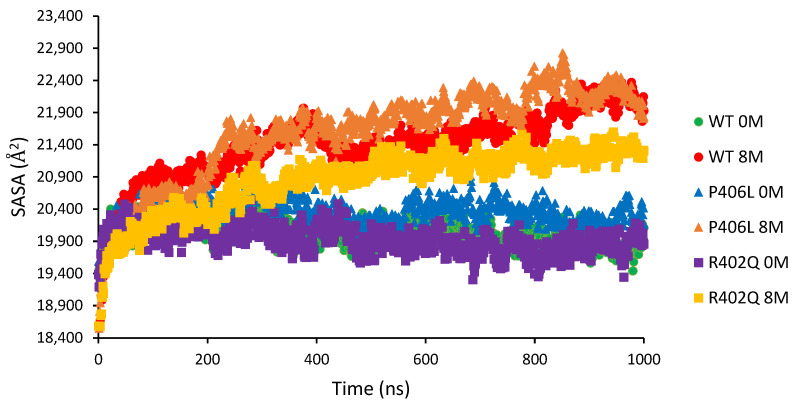
Comparing MD trajectories between wild type and mutants for water and 8M urea environments. Displayed here are the average SASA values for the WT (circles), P406L mutant (triangles), and R402Q mutant (squares) in water and 0M urea at 1 μs. Each condition was run in triplicate from which the averages were found.

**Figure 5 ijms-24-13032-f005:**
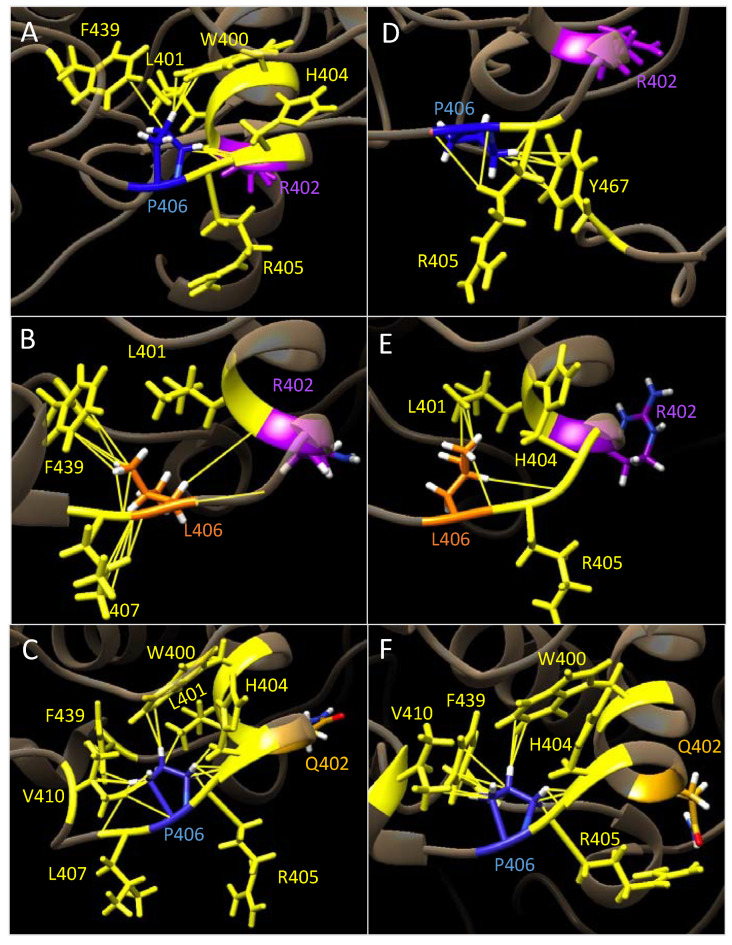
Interatomic hydrophobic contacts with residue 406 at 1 μs. Residue P406 is colored blue and R402 is purple. Substitutions P406L and R402 are orange and gold, respectively. Residues contacting position 406 along with the specific contacts between them are colored yellow. Panel (**A**): Contacts of WT in water. Panel (**B**): P406L in water. Panel (**C**): R402Q in water. Panel (**D**): WT in 8M urea. Panel (**E**): P406L in 8M urea. Panel (**F**): R402Q in 8M urea.

**Figure 6 ijms-24-13032-f006:**
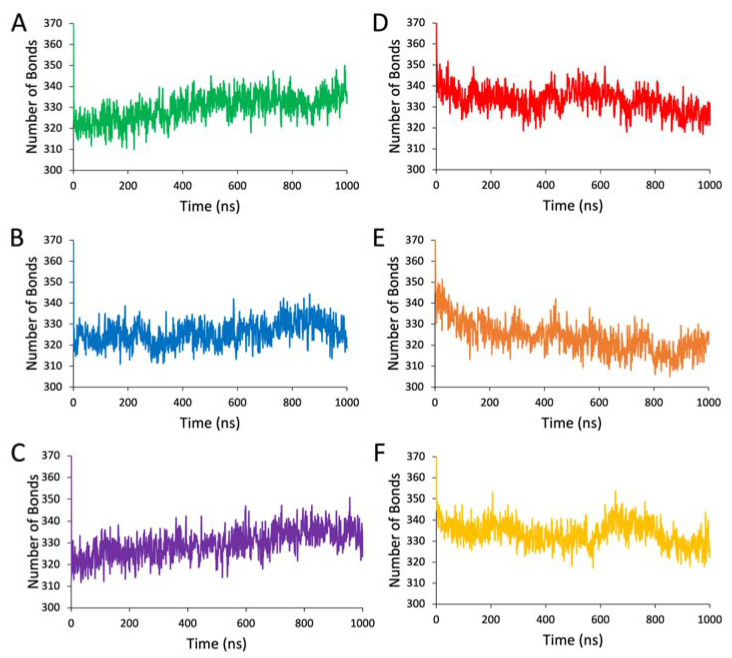
Number of intramolecular hydrogen bonds over time (ns). Panel (**A**): WT Tyr in water. Panel (**B**): The P406L variant in water. Panel (**C**): The R402Q variant in water. Panel (**D**): The WT in 8M urea. Panel (**E**): P406L variant in 8M urea. Panel (**F**): R402Q variant in 8M urea.

**Figure 7 ijms-24-13032-f007:**
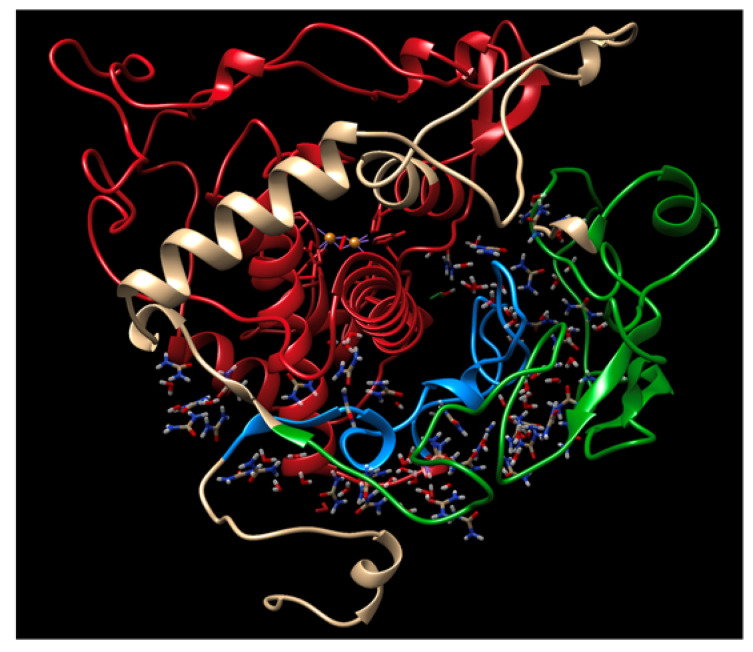
Solvent within the cleft of the subdomain interface and cysteine-rich region. The active site of Tyr lies within the catalytic Tyr subdomain (red). The cysteine-rich subdomain rests on the periphery of the globule (green). The residues beyond the catalytic region make up the subdomain interface (blue).

**Figure 8 ijms-24-13032-f008:**
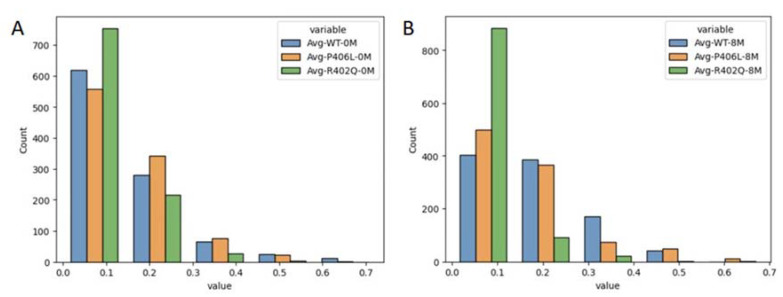
Ionic interaction energy distributions at the interface of the cysteine-rich subdomain and globular C-terminal region with time. Panel (**A**): Distributions for all proteins in water. Panel (**B**): Distributions for all proteins in 8M urea.

**Figure 9 ijms-24-13032-f009:**
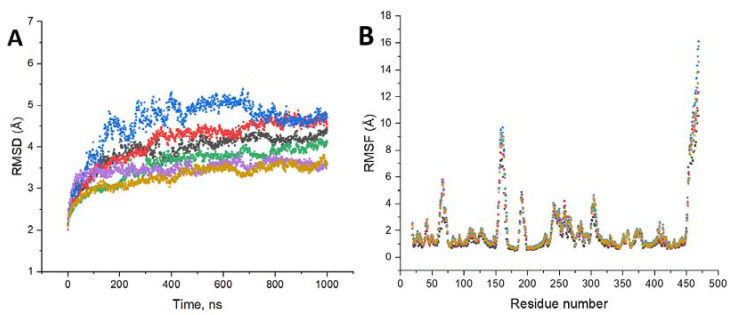
Results of root mean square deviations (RMSD) and root mean square fluctuations (RMSF) calculations. Panel (**A**): RMSD plots were obtained by averaging over the triplicates of independent runs. Here are wild-type proteins at 0M (black squares) and 8M (red circles), P406L mutant variant at 0M (blue triangle) and 8M (green triangle), and R402Q mutant variant at 0M (light purple rhomb) and 8M (beige triangle) urea concentrations. Panel (**B**): RMSF plots show that the loop regions and C-terminus RMSF values are larger compared to those of regular secondary structures such as helix and beta-sheet conformations. Here are wild-type proteins at 0M (black squares) and 8M (red circles), P406L mutant variant at 0M (blue triangle) and 8M (green triangle), and R402Q mutant variant at 0M (light purple rhomb) and 8M (beige triangle) urea concentrations.

**Table 1 ijms-24-13032-t001:** OCA1B mutations and unfolding mutagenesis screen data. accessed on 1 May 2022.

Mutation	Conditions	Clinical Significance	Unfolding Fraction	Foldability	ΔΔG, Kcal/mol
P406L	OCA1(A/B)	P/LP	1	15	4.27
R402Q	OCA1(B)	CIP	0.97	11	2.08

Conditions and clinical significance data were retrieved from the ClinVar database. P/LP: Pathogenic/likely pathogenic. CIP: Conflicting interpretations of pathogenicity.

**Table 2 ijms-24-13032-t002:** Total cavity volumes increase after 1 μs of MD in water and 8M urea.

Structure	Cavity Volume, Å^3^	
	Water	8M Urea
WT	20,556 ± 1633	24,992 ± 912
P406L	19,842 ± 2265	24,686 ± 735
R402Q	20,178 ± 1394	24,139 ± 2054

**Table 3 ijms-24-13032-t003:** Linear fitted change in intramolecular hydrogen bonds over 1 μs.

Structure	ΔHydrogen Bonds in Water	ΔHydrogen Bonds in 8M Urea
WT	13	−9
P406L	8	−18
R402Q	14	−6

ΔHydrogen bonds in water were taken by subtracting the 1000 ns and 0 ns values from the linear fit equation and rounded to the nearest whole number.

## Data Availability

The TYR homology model and global mutagenesis data, and molecular dynamics simulations for TYR and mutant variants are freely available at the NEI Commons website (https://neicommons.nei.nih.gov/#/proteomeData, accessed on 14 February 2023).

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
