# Peer review of "Evaluating the Cysteine-Rich and Catalytic Subdomains of Human Tyrosinase and OCA1-Related Mutants Using 1 μs Molecular Dynamics Simulation"

_ijms, 2023, doi:10.3390/ijms241713032_

Round 1

Reviewer 1 Report

Woods et al. provided a computational study to characterize the unfolding process of Tyr, a protein involved in melanogenesis pathway.

The Authors performed 6 MD simulations of 1 us each, of the WT protein as well as of two mutants (P460L and R402Q) in water and in urea (with a concentration of 8M).

Analyzing the trajectories they discovered that the mutation in P406L decreases the rigidity of the catalytic subdomain, while the mutation in R402Q introduces local rigidity.

I think that the manuscript is well presented and with a good workflow.

Anyway, I suggest some major revisions that will improve the quality of the paper.

Major:

The only thing that is missing in this paper is a principal component analysis (PCA) computed on the carbon alpha of the proteins. The analysis should compute on the concatenated trajectories to see how much the conformations sampled are similar or not. Maybe the Authors can divide the analysis in two, one for the systems in water and one for the systems in urea. This will help to see the changes in the conformations explored by the proteins. If the Authors are not confident with this analysis, I suggest a couple of papers concerning the PCA (https://doi.org/10.1002/prot.340170408; https://doi.org/10.1371/journal.pone.0121114). 

Minor:

  1. In the abstract the Authors should underline that they performed 6 simulations, as this is not clear until the results section. Considering that this is a very good point, I would suggest to stress it more.

  2. Concerning the graph of the interaction energy (Figure 8), I would suggest providing an histogram instead of a distribution. This will allow to immediately visualize differences in energy values. Maybe, the Authors can divide the analysis in two,  thus plotting the systems in water on a graph, and the systems in urea on another one.

  3. I would suggest reporting the graphs concerning basic structural analysis as RMSD and RMSF. This will be useful to see convergence of the proteins in the different conditions and also the residues that fluctuate more.

Best,

Author Response

  1. In the abstract the Authors should underline that they performed 6 simulations, as this is not clear until the results section. Considering that this is a very good point, I would suggest to stress it more.

We stressed in an abstract that 6 simulations were performed. Here is a new version of abstract:

‘In protein unfolding, the roles of these subdomains are not well established. Here, we performed 6     molecular dynamics simulations at room temperature for Tyr and OCA1-related mutant variants P406L and R402Q intra-melanosomal domains. The proteins were simulated for 1 μs in water and urea to induce unfolding.’  

  1. Concerning the graph of the interaction energy (Figure 8), I would suggest providing an histogram instead of a distribution. This will allow to immediately visualize differences in energy values. Maybe, the Authors can divide the analysis in two, thus plotting the systems in water on a graph, and the systems in urea on another one.

A new Figure 8 containing histogram was prepared as requested by the reviewer.

Figure 8: Ionic interaction energy distributions at the interface of the cysteine-rich subdomain and globular C-terminal region with time. Panel A: Distributions for all proteins in water. Panel B: Distributions for all proteins in 8M urea.

  1. I would suggest reporting the graphs concerning basic structural analysis as RMSD and This will be useful to see convergence of the proteins in the different conditions and also the residues that fluctuate more.

The only thing that is missing in this paper is a principal component analysis (PCA) computed on the carbon alpha of the proteins. The analysis should compute on the concatenated trajectories to see how much the conformations sampled are similar or not. Maybe the Authors can divide the analysis in two, one for the systems in water and one for the systems in urea. This will help to see the changes in the conformations explored by the proteins. If the Authors are not confident with this analysis, I suggest a couple of papers concerning the PCA (https://doi.org/10.1002/prot.340170408; https://doi.org/10.1371/journal.pone.0121114). 

.           This request is a very reasonable suggestion from the Reviewer. Unfortunately, standard PCA programs we were trying to use for the analysis such as Excel and OriginLab are limited in the number of data points and were failing and cannot hold ~50% of the data necessary for PCA analysis. In addition, all calculations of rmsd and rmsf should be performed for triplicates of 3 proteins. In our work, we are using the Yasara package. This program does not have any options for this kind of calculations. This means that we must use another software or write a code to satisfy the Reviewers, which could not be performed in a 10- or 20-days period. We are going to use this kind of analysis in the future but right now it is simply not possible.

Reviewer 2 Report

Attached file

Author Response

Reviewer 2

 (i) Which PDB was selected as a template for homology modeling?

The template of protein is based off of the Tyrp1 crystal structure 5M8L.    

(ii) How pKa calculations were carried out (e.g. using the PROPKA method)?;

The cell neutralization and pKa prediction script within Yasara was used to predict pKa values as a function of electrostatic potentials calculated by Ewald summation, hydrogen bonds and accessible surface areas of titratable groups.

(iii) MD systems: it should be more detailed: which solvent models (urea and water), Minimizations, heating, and equilibration procedures? How to probe that all systems were well equilibrated?

            All proteins were simulated in a cubic cell. Solvent environments were energy minimized before each protein structure was added, and the completed simulation system was energy minimized again before beginning each MD experiment. We evaluated protein equilibration using the changes in the accessible surface of each globule. This was shown by sharp changes in surface area values followed by a flattening curve which shows the proteins have adjusted to the simulation environments. This flattening curve was observed in all structures within the first 100 ns of the simulations.

(iv) More critical point: which model (parameters) was used to simulate Cu-His coordination? I recommend these references: J. Chem. Theory Comput., 2010, 6, 2935-2947; J. Phys. Chem. Lett. 2015, 6, 2657–2662; Int. J. Mol. Sci. 2020, 21, 4783.

            Yasara is unable to simulate coordinate covalent bonds, so in replacement of this, pseudo bonds are added between these atoms which maintain their specified bond lengths and angles throughout the simulations. Charges for the Cu ions were automatically assigned by the Yasara AutoSMILES program.    

2) RMSD and RMSF are good tools for structural analysis (as well as other computational techniques applied here), I strongly recommend running PCA and FEL analysis to improve enzymatic stabilization from structural and thermodynamic perspectives.

This request is a very reasonable suggestion from the Reviewer. Unfortunately, standard PCA programs we were trying to use for the analysis such as Excel and OriginLab are limited in the number of data points and were failing and cannot hold ~50% of the data necessary for PCA analysis. In addition, all calculations of rmsd and rmsf should be performed for triplicates of 3 proteins. In our work, we are using the Yasara package. This program does not have any options for this kind of calculations. This means that we have to use another software or write a code to satisfy the Reviewers, which could not be performed in a 10- or 20-days period. We are going to use this kind of analysis in the future but right now it is simply not possible.

3) The quality (resolution and description) of the Figures should be improved.

The resolution of Figures has improved.

In addition, Figure 5 caption was changed:

Inter-atomic hydrophobic contacts with residue 406 at 1 μs. Residue P406 is colored in blue and R402 is colored in purple. Substitutions P406L and R402 are colored in orange and gold, respectively. Residues contacting position 406 along with the specific contacts between them are colored in yellow. Panel A: Contacts of WT in water. Panel B: P406L in water. Panel C: R402Q in water. Panel D: WT in 8M urea. Panel E: P406L in 8M urea. Panel F: R402Q in 8M urea.

  •  

Round 2

Reviewer 1 Report

Dear Authors,

As Molecular dynamics (MD) is a computational technique, not reporting data because of the inadequacy of the Software cannot be accepted. There is a reason why all most all the computational researchers use well known Software as GROMACS or NAMD that are all FREE of charge. I can make an effort to understand problems in performing a PCA, but RMSD and RMSF are standard analyses that the Authors should have provided before submitting the manuscript. In fact these analyses are at the bases of MD. How the Authors and the readers could know the convergences of the systems if nobody has checked them? Normally these analyses take 1 day to be performed, for all the replicas. To make a practical example, it is like the Authors are growing cells without checking the optical density of the medium, thus the cells can be all dead or not grown up enough.

Moreover, Yasara provided somehow the possibility to compute the RMSF (http://www.yasara.org/mdanalysis.htm). Finally, as we are talking about science, analysis must be reported for all the copies, as Authors must provide controls. If the Authors were working in vitro they should have provided analysis and controls for all the samples. MD is the same. The fact that it is a computational technique does not mean that it should overpass the bases of research and science.

I am sorry but I cannot accept the paper in its current state.

At least RMSD and RMSF must be provided for all the replicas. Otherwise the Authors should try to publish their manuscript in another Journal.

Author Response

We are very thankful to Reviewer 1 for this comment. We calculated RMDs and RMSFs, as described in Methods section 4.7.

‘4.7. Measures For MD Quality Assessment

The RMSD of an ensemble of structures from a reference structure was calculated using the RMSD command in Yasara and additional in-house Python scripts. The ensemble of molecular dynamics snapshots of 1 ns interval was aligned with the reference structure, based on a set of selected Ca atoms followed by calculations of RMSD values between corresponding atoms of the snapshots and reference structure.

The RMSF was generated using the superimposed molecular dynamics snapshots for every 25 ns interval. These superposed structures were saved as PDB files with their updated coordinates to remove all translational and rotational movements. The RMSF analysis was carried out by loading each superposed structure into Yasara, adding the Ca-atom positions to a calculation table using the AddPosAtom function to calculate the averages and standard deviations using the built-in RMSF command. ‘

We also added Figure 9 (Panels A, B) with a caption and a small paragraph (2.7) in the Results section.

‘2.7. RMSD and RMSF plots.

Calculations for RMSDs and Ca-RMSFs were performed using Yasara as described in the Methods section. RMSDs were generated by superposing each molecular dynamics snapshot to their Ca atoms to the original structure, calculated, and plotted in Figure 9 (Panel A). These superposed structures were saved as PDB files for the subsequent RMSF analysis and plotted in Figure 9 (Panel B). Interestingly, the RMSD values showed the highest deviations in P406L in water and the wild type in 8M urea. Similarly, to our SASA calculations, the R402Q structure retains its structure better than all other proteins. During the MD runs the most significant fluctuations were observed for the loop formed by residues 150-170 and for the C-terminus located above the residue 450. Contrary to the RMSD values, there is much overlap between RMSF peaks across all structures with only the wild type in water displaying reduced fluctuations. These reduced fluctuations are observed at the peaks between residues 50-75, 150-170, and to a lesser extent 300-315 and 400-420.

Figure 9. Results of root mean square deviations (RMSD) and root mean square fluctuations (RMSF) calculations. Panel A: RMSD plots were obtained by averaging over the triplicates of independent runs. Here are wild-type proteins at 0M (black squares) and 8M (red circles), P406L mutant variant at 0M (blue triangle) and 8M (green triangle), and R402Q mutant variant at 0M (light purple rhomb) and 8M (beige triangle) urea concentrations. Panel B: RMSF plots show that for the loop regions and C-terminus RMSF values are larger compared to that of regular secondary structures such as helix and beta-sheet conformations. Here are wild-type proteins at 0M (black squares) and 8M (red circles), P406L mutant variant at 0M (blue triangle) and 8M (green triangle), and R402Q mutant variant at 0M (light purple rhomb) and 8M (beige triangle) urea concentrations.’

Hopefully, now we satisfy the Reviewer 1 request.

Reviewer 2 Report

Dear Editor,

I would like to congratulate the authors on their revised version of manuscript entitled “Evaluating the Cysteine-Rich and Catalytic Subdomains of Human Tyrosinase and OCA1-related Mutants using 1 us Molecular Dynamics Simulation” (Manuscript ID: ijms-2453200-v2).

In this version most of my previous concerns were attended by authors. I have just one suggestion before publication in the IJMS:

1)    In Figure 6, the linear equation is unnecessary. To remove fitted line.

High quality of English.  

Author Response

  1. I would like to congratulate the authors on their revised version of manuscript entitled “Evaluating the Cysteine-Rich and Catalytic Subdomains of Human Tyrosinase and OCA1-related Mutants using 1 us Molecular Dynamics Simulation” (Manuscript ID: ijms-2453200-v2).

We are very thankful to Reviewer 2 for this comment.

  1. I have just one suggestion before publication in the IJMS: In Figure 6, the linear equation is unnecessary. To remove the fitted line.

The fitted line in Figure 6 has been removed. 

Round 3

Reviewer 1 Report

Dear Authors,

I appreciated your effort, so I decided to accept the manuscript.

Anyway, I would like to suggest you to switch to other MD simulation software that allows to perform easy calculations, such as RMSD and RMSF, without writing any code. In fact, these software have been around for 30-20 years and they are well-tested and trusted.

Kind regards,